# Quantifying the Latency and Possible Throughput of External Interrupts on Cyber-Physical Systems

Oliver Horst
Johannes Wiesböck
fortiss GmbH
Research Institute of the Free State of Bavaria
Munich, Germany
{horst,wiesboeck}@fortiss.org

Raphael Wild
Uwe Baumgarten
Technical University of Munich
Department of Informatics
Garching, Germany
{raphael.wild,baumgaru}@tum.de

## ABSTRACT

An important characteristic of cyber-physical systems is their capability to respond, in-time, to events from their physical environment. However, to the best of our knowledge there exists no benchmark for assessing and comparing the interrupt handling performance of different software stacks. Hence, we present a flexible evaluation method for measuring the interrupt latency and throughput on ARMv8-A based platforms. We define and validate seven test-cases that stress individual parts of the overall process and combine them to three benchmark functions that provoke the minimal and maximal interrupt latency, and maximal interrupt throughput.

## DATA AVAILABILITY STATEMENT

A snapshot of the exact version of the prototyping platform toki [12] that was used to conduct the presented measurements is available on Zenodo [13]. The snapshot also contains the captured, raw STM trace data and scripts to produce the presented figures. The latest version of toki can be obtained from [10].

## 1 INTRODUCTION

Cyber-physical systems (CPSs) are characterized by the fact that a computer system works together with a physical environment, or rather controls it. A specific characteristic of such control systems is their necessity to provide short and predictable reaction times on events in the physical world, to guarantee a good control quality [14]. Both properties are likewise essential for modern systems, such as tele-operated-driving [27], and classic systems, such as the control of internal combustion engines [11].

An important aspect of the achievable reaction time is the interrupt handling performance in both dimensions the interrupt handling latency and throughput capabilities of a system. Especially the effect of the utilized software stack has not yet been comprehensively assessed. Such a systematic evaluation would, however, facilitate the development and selection of CPS software stacks for particularly latency-sensitive or throughput-hungry use-cases.

Previous studies in this field mainly conducted measurements with the help of external measurement devices [16, 19, 22, 26],

which requires an in-depth understanding of the hardware to obtain precise measurements [22]. This expert knowledge, however, is reserved for the system on chip (SoC) and processor intellectual property (IP) vendors. Hence, we see the need for a measurement method that allows to accurately measure and properly stress the interrupt handling process of today's SoCs without expert knowledge. Accordingly, we present a flexible interrupt performance measurement method that can be applied to ARMv8-A IP-core based platforms that provide a physical trace-port. As we see an increasing market share of ARM based systems [20] and their wide adoption in automotive CPSs [9, 25, 28] we strongly believe that our method helps in analyzing a multitude of relevant systems.

We specify three benchmark functions based on the assessment of ten combinations out of seven distinctive test-cases. Whereby each test case was chosen to stress a dedicated part of the ARM interrupt handling process. The effectiveness of the test-cases and benchmark functions is demonstrated on a Xilinx ZCU102 evaluation board [30] with two different software stacks.

In summary, we contribute *(i)* a precise method to measure the interrupt performance of complex ARM based SoCs without expert knowledge, and *(ii)* a set of benchmark functions that provokes the best and worst interrupt latency and maximal throughput on a given ARMv8-A hardware and software combination.

The rest of this paper is structured as follows: Section 2 describes the interrupt handling process on ARMv8-A platforms with a Generic Interrupt Controller (GIC) version 2, Section 3 presents the measurement setup and procedure of the envisioned evaluation method, Section 4 discusses the proposed test-cases and benchmarks along with the measurement results, Section 5 gives an overview on related work, and Section 6 concludes the paper.

## 2 INTERRUPT HANDLING PROCEDURE ON ARMv8-A PLATFORMS

Müller and Paul [21] define an interrupt as an event that causes a change in the execution flow of a program sequence other than a branch instruction. Its handling process starts with the activation through a stimulus and ends with the completion of the interrupt service routine (ISR), which is called in consequence and processes the stimulus. Until the ISR is executed several steps are undergone in hardware to cope for example with simultaneously arriving interrupt requests (IRQs) and masking of certain requests. In the following, we explain this process for ARMv8-A platforms, as specified in the GIC architecture specification version 2 [3]. In Section 4 this information is used to design suitable test cases.

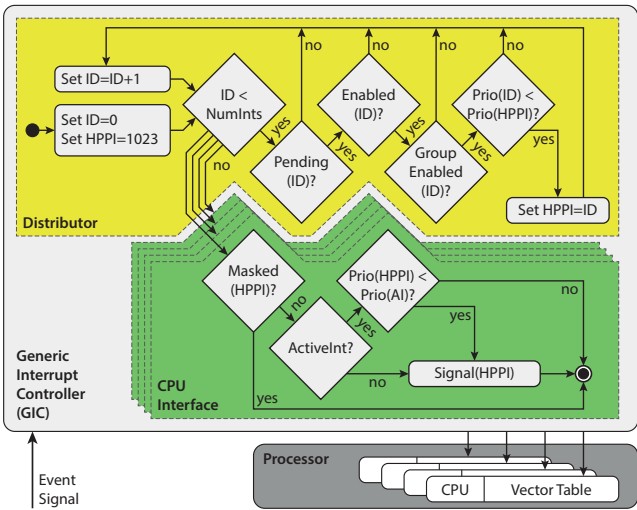

**Figure 1: Repeatedly, per CPU interface executed selection and signaling process of the Generic Interrupt Controller (GIC) for handling triggered interrupt requests (IRQs).**

The GIC architecture specification differentiates among four types of interrupts: peripheral, software-generated, virtual, and maintenance interrupts. In the course of this paper we focus solely on measuring interrupts triggered by external stimuli, the peripheral interrupts. They can be configured as edge-triggered or level-sensitive. This means that the corresponding interrupt is recognized either once on a rising edge in the input event signal, or continuously as long as the signal has a certain strength.

The GIC supervises the overall interrupt routing and management process up to the point that the ISR is called. Figure 1 shows the GIC architecture and the signaling path for the selected measurement hardware, the Xilinx UltraScale+ MPSoC (ZynqMP). The GIC manages all incoming event signals of the system and consists out of a central Distributor and a CPU interface per processor core. The Distributor manages the trigger-type of each interrupt, organizes their prioritization, and forwards requests to the responsible CPU interface. The CPU interfaces perform the priority masking and preemption handling for their associated core.

The timeline of the various steps in the interrupt handling process are illustrated in Fig. 2. The handling process of a certain IRQ begins with the arrival of an event signal at the Distributor (step 1). In case the signal matches the configured trigger type, the actual handling process is triggered through the recognition of a new IRQ (step 2), which eventually leads to the execution of the ISR (step 9).

After being recognized (step 2), the Distributor may select and forward the IRQ to the responsible CPU interface according to the process depicted in the upper half of Fig. 1. This selection process (step 3) is executed repeatedly and potentially in parallel for each CPU interface. When the next highest priority pending interrupt (HPPI) is identified the Distributor forwards the request to the currently inspected CPU interface (step 4). The CPU interface filters incoming requests according its configuration and the process shown in the lower half of Fig. 1. As a result the CPU interface may signal a pending IRQ to its associated core (step 5).

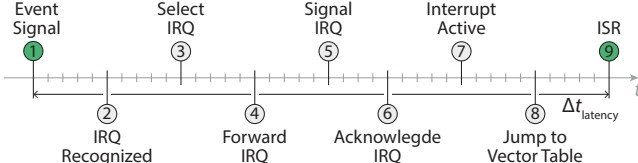

**Figure 2: Overall steps of the interrupt handling process on ARMv8 platforms with a Generic Interrupt Controller (GIC) version 2 and an ARMv8-A architecture profile core.**

Subsequent to the signaling of a new IRQ, on an ARMv8-A architecture [6], the core acknowledges the signaled IRQ by reading its id (step 6), which marks the IRQ as *active* within the GIC (step 7). Meanwhile, the processing core jumps to the vector table entry indicated by the id of the current IRQ (step 8), which leads to the execution of the ISR (step 9). Individual software stacks might add additional steps before the ISR finally starts.

Besides the regular interrupt requests described above, the GIC architecture version 2 (GICv2) also supports the prioritized and more secure fast interrupt requests (FIQs), however, these lay out of the focus of this paper. Furthermore, it shall be noted that the regular interrupt processing path was not affected by the latest updates to the GIC architecture (*i.e.*, version 3 and 4). Hence, we strongly believe that the presented benchmark functions would be still valid for the updated architectures.

## 3 PROPOSED EVALUATION METHOD

With respect to the interrupt performance of a hardware/software combination two properties are most relevant: the IRQ *latency* and *throughput*. Considering the process shown in Fig. 2, we define the IRQ *latency* as the time between a signaled event (step 1) and the call to the corresponding ISR (step 9). As *throughput* we understand the number of completed ISRs per time unit.

### 3.1 Measurement Setup

Our measurements utilize a Xilinx ZCU102 [30], with the toki prototyping platform [10, 12], and an ARM DSTREAM [5] hardware trace. We have chosen the ZynqMP, as it features an in-built programmable logic (PL) and versatile tracing capabilities. Figure 3 illustrates the chosen hardware setup. The ZynqMP is divided into the PL and processing system (PS). The PS provides two processor clusters, the application processing unit (APU) with four ARM Cortex-A53 cores and the real-time processing unit (RPU) with two ARM Cortex-R5 cores. Both clusters have their own interrupt handling infrastructure, but we focus on the APU's only.

The ARM CoreSight [4] tracing capabilities of the ZynqMP allow to record events, such as taken branches, external interrupts, and software events on a common timeline defined by system-wide timestamps. The system trace macrocell (STM) and embedded trace macrocells (ETMs) record the various events in hardware, without altering the behavior of the executed code, neither in a temporal nor semantic sense. Only software driven events require a register write operation and thus marginally influence the timing of the executed code. CoreSight is part of all newer ARM processor IPs and can be utilized as soon as the used hardware features a physical

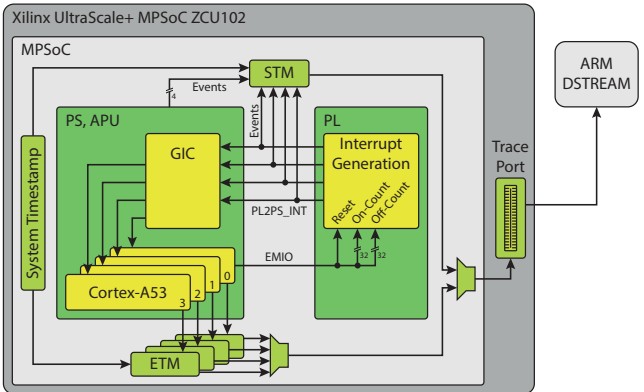

**Figure 3: Chosen measurement setup, with four PL2PS interrupts generated by the programmable logic (PL) according to the configuration signaled by the processing system (PS) via its extended multiplexed I/O interface (EMIO). The generated interrupts, executed instructions, and global timestamps are recorded through the system trace macrocell (STM) and embedded trace macrocell (ETM). The captured trace is read via an ARM DSTREAM unit.**

JTAG- and trace-port. The latter is typically available on evaluation boards used for prototyping professional applications.

In addition to the in-built hardware features, we deploy a custom interrupt generation block into the PL. The block allows to simultaneously stimulate APU interrupts, following a generation pattern defined by a logical-high and -low phase, and trace them in the STM. This could also be realized with an external field-programmable gate array (FPGA) or a signal generator, to support additional platforms. The pin multiplexing configuration of the target platform only has to ensure that triggered input lines are also connected to a STM hardware event channel.

### 3.2 Measurement Procedure

Based on the measurement setup, we propose two measurement procedures, one per measurement type (*i.e.*, latency and throughput), utilizing two different configurations of the interrupt generation block. Both procedures use APU core 0 to take measurements and the other cores as stimulation sources, where needed.

We conduct our measurements on two software stacks: *(i)* a baremetal, and *(ii)* a FreeRTOS based system. Both software stacks are provided by the toki build- and test-platform [10, 12], and utilize a driver library provided by Xilinx [32]. This library already features an interrupt dispatching routine that multiplexes the processor exception associated with regular interrupts on the target.

The bare-metal stack is an unoptimized piece of code that features neither a scheduler, nor a timer tick. It could clearly be optimized, however, it is unclear how much a fully optimized assembler version of the stack would impact the presented results.

Thanks to its Yocto [33] based build-system, toki can easily be executed to include Linux based software stacks and with that the presented test setup. Completely different software stacks and hardware platforms can be evaluated with the given setup, when they

provide *(i)* a libc function interface, and *(ii)* drivers for interacting with the caches, STM trace, and GIC on that platform.

*Throughput.* In case of a throughput evaluation, we configure the interrupt generation block to continuously signal the PL2PS interrupts for 9.75 s and then wait for another 250 ms on a rotating basis. We call each repetition of this pattern a stimulation phase. Core 0 is configured to handle the signaled private peripheral interrupt (PPI) on a level-sensitive basis and the corresponding ISR does nothing despite emitting a STM software event, *i.e.*, executing a 8 bit store instruction. Hence, the time spent in each ISR and with that the possible reduction of the maximum throughput is negligible.

For each throughput measurement we capture a 120 s trace and evaluate the contained stimulation phases. The throughput ($\mu$) in each stimulation phase ($i \in [0, 19]$) is obtained from the traced measurement samples by counting the ISR generated STM-software-events between the rising-edge STM-hardware-event of the ($i$)$^{th}$ and ($i+1$)$^{th}$ stimulation phase and dividing it with the length of one logical-high-phase. The set of throughput values considered for the evaluation in Section 4 is then given by $M = \{\, \mu(i) \mid i \in [0, 19] \,\}$.

*Latency.* The latency evaluation is conducted with an alternating scheme of a 1 ms PL2PS interrupt trigger and a 4 ms pause. The interrupt generation block is configured accordingly. Again we refer to each repetition of this scheme as a stimulation phase. In contrary to the throughput measurement, however, core 0 is configured to handle the signaled PPI on a rising-edge basis. Thus, every stimulation phase provokes only one interrupt. The corresponding ISR is the same as for the throughput measurements.

The results for the latency measurements are obtained by evaluating 30 s trace captures. The interrupt latency $\Delta t_{\text{latency}}(i)$ induced by each stimulation phase $i \in [0, 2399]$ is given by $\Delta t_{\text{latency}} = B - A$, with $A$ representing the point in time where the interrupt was stimulated and $B$ the point where the corresponding ISR was started. Both points can be obtained from the captured trace. $A$ is given by the timestamp of the STM hardware event associated with the rising-edge of the PL2PS interrupt signal. $B$, on the other hand, has to be determined and defined for every analyzed software stack individually. In the course of this paper we utilize the timestamp of a STM event generated within the interrupt handler of our benchmark application that runs on top of the evaluated software stacks.

Similar to the throughput values, the set of latency values considered in Section 4 is given by $X = \{\, \Delta t_{\text{latency}}(i) \mid i \in [0, 2399] \,\}$.

### 3.3 Precision and Limitations

In our measurement setup, we configure the PL, trace port, and timestamp generation clock to oscillate at 250 MHz. Hence, two consecutive timestamp ticks lay 4 ns apart from each other. Since each sampled event in the ETM and STM is assigned a timestamp, our measurement precision corresponds exactly to the system timestamp resolution, *i.e.*, 4 ns. This is an order of magnitude smaller than the interrupt latency measured in a previous study for the same hardware platform [29] and a quarter of the measured minimal interrupt latency of an ARM real-time core [22].

Even though state of the art oscilloscopes provide a sampling rate of up to 20 GSa/s [15], which corresponds to a measuring precision of 0.05 ns, the actual precision in case of interrupt latency

measurements might be considerably lower. The reason for this is that the oscilloscope can only measure external signals of a processor. Thus, in-depth knowledge of the internal structure of the hardware and executed instructions during a measurement is required to utilize the full precision of the oscilloscope. This makes it less suited for the evaluation of different hardware platforms and software stacks. The CoreSight based measurement setup, on the other hand, supports a flexible placement of the measurement points within and outside of the processor and does not require any expert knowledge about the hardware or software.

Besides the measurement precision and flexibility, we also need to ensure that the presented measurement setup is sane and triggered interrupts can actually be recognized by the processor. According to the ZynqMP technical reference manual [31, p. 312], a signal pulse that shall trigger a PL2PS interrupt needs to be at least 40 ns wide to guarantee that it is recognized as such. Hence, the presented stimulation scenarios for the two measurement procedures ensure that all triggered interrupts can be recognized.

The disadvantage of the presented measurement approach, however, is that it is only applicable for ARM based platforms with a dedicated JTAG- and trace-port. Given ARM's 40% share of the semiconductor market for IP designs [20] and the wide availability of suitable evaluation boards, we believe this is an acceptable drawback. An additional limitation is that valid measurements can only be obtained for the interrupt with the highest priority among the active ones, but this applies to any kind of measurement setup.

## 4 CONSTRUCTING A BENCHMARK

In order to create a benchmark for comparing the interrupt latency and throughput across platforms and software stacks, we have designed seven test-cases specifically tailored to stress the ARMv8-A interrupt handling process. To judge their suitability for an overall benchmark, we measure their performance with the two software stacks described in Section 3.2 on top of the ZynqMP discussed in Section 3.1. By comparing the impact of each test-case with respect to the baseline performance of the two systems, we compose three benchmarks out of the test-cases and show their suitability by applying them to the same system configurations.

### 4.1 Evaluated Test-Cases

Given the interrupt handling process in Section 2, we conclude that the time spent in the process can be influenced by: the core, caches, memory, and GIC. We have designed seven test-cases that aim to reveal the influence of different configuration settings related to the aforementioned components onto the temporal behavior of the interrupt handling process. However, we do exclude the core from our considerations by only measuring interrupts with the highest priority and not computationally loading the measured core. The measurements for all test-cases follow the scheme presented in Section 3.2, unless indicated otherwise. Depending on the goal of each test-case they are either applied only for latency measurements or both latency and throughput measurements. The proposed test-cases and their targeted components are summarized in Table 1, and Figs. 4 and 5 present the results of our measurements. The presented results are based on 848–6000 measurement samples per latency measurement and 11–12 samples per throughput measurement.

The remainder of this section elaborates on the intended influence of the listed test-cases on the interrupt handling process.

*T1: Baseline.* T1 is intended to provide a reference point to compare the other test-cases to and rate their impact. Hence, T1 assess the interrupt latency and throughput of a system in the most isolated way, with only one core and interrupt enabled and caches disabled. Hence, T1 only enables the extended multiplexed I/O interface (EMIO) pin driven interrupt and routes it to core 0. As ISR the default handler, described in Section 3.2, is used. T1 is evaluated for its latency and throughput performance.

*T2: Caches enabled.* T2 equals T1, with the exception that all operations are executed with enabled caches. This test is conducted for both latency and throughput measurements.

*T3: Caches invalidated.* T3 is also based on T1, but the ISR additionally invalidates the data and instruction caches. Due to the fact that this is not feasible in throughput measurements, as new interrupts would arrive independently of the cache invalidation process, we conduct only latency measurements with T3.

*T4: Enabled interrupts.* T4 aims at stressing the GIC with the highest possible number of enabled interrupts, as the interrupt selection and signaling process suggests that more checks have to be done the more interrupts are enabled/pending. Hence, T4 enables the maximum number of interrupts supported by the ZynqMP, except those required for conducting the measurements. All interrupts are routed to and handled by core 0. The measured PL-to-PS interrupt is assigned to the highest priority and all other interrupts to the lowest priority. Core 0 installs an empty ISR that immediately returns after clearing the IRQ in the GIC for all interrupts, except the measured PL-to-PS interrupt, which uses the same ISR as T1.

As this test aims at stressing the GIC to reduce its performance, we only evaluate it with respect to the interrupt latency. To be able to identify trends, we evaluated this test-case with 1, 36, 72, 108, 144, and 180 stressing interrupts. However, due to the marginal differences between the results of the different T4 variants and space constraints we only show the results of T4-180, T4 with 180 stressing interrupts, which provoked the highest latency.

*T5: Order of priorities.* T5 utilizes the same setup as T4 and is also applied to latency measurements only. However, in contrast to T4, T5 only utilizes as much interrupts as there are priorities, *i.e.*, 15. The measured interrupt remains at priority 0 and the priorities of the other 14 are assigned in an ascending order (*i.e.*, 14 to 1). This design intends to provoke a maximal number of HPPI updates.

*T6: Parallel interrupt handling.* To test the influence of parallelly handled interrupts on the interrupt handling process, T6 enables up to 4 cores and configures all of them to handle the EMIO pin 0 interrupt. The interrupt is configured as level-sensitive with the highest priority. The PL ensures that this interrupt is signaled continuously and simultaneously as soon as the test is enabled. The ISRs on all cores generate a STM event, which are evaluated for throughput measurements. In case of latency measurements, however, only those STM events produced by core 0 are considered.

We evaluated T6 with 2, 3, and 4 enabled cores. The results showed a clear trend that the more enabled cores the higher the observed latency and the lower the achieved throughput. Due to

**Table 1: Properties of the evaluated test-cases and benchmarks used to compare the interrupt latency (L) and throughput (T).**

| Description | Targeted Component | Measurements | | Enabled Interrupts | Cache Config | Enabled Cores | Benchmarks | | |
|---|---|---|---|---|---|---|---|---|---|
| | | L | T | | | | $L_{min}$ | $L_{max}$ | $T_{max}$ |
| T1: Baseline | — | X | X | 1 | Disabled | 1 | | | |
| T2: Caches enabled | Cache | X | X | 1 | Enabled | 1 | X | | X |
| T3: Caches invalidated | Cache | X | | 1 | Invalidated | 1 | | | |
| T4: Enabled interrupts | GIC | X | | 2–181 | Disabled | 2 | | X | |
| T5: Order of priorities | GIC | X | | 15 | Disabled | 2 | | | |
| T6: Parallel interrupt handling | GIC | X | X | 1 | Disabled | 2, 3, 4 | | X | X |
| T7: Random memory accesses | Memory | X | | 1 | Disabled | 4 | | | |

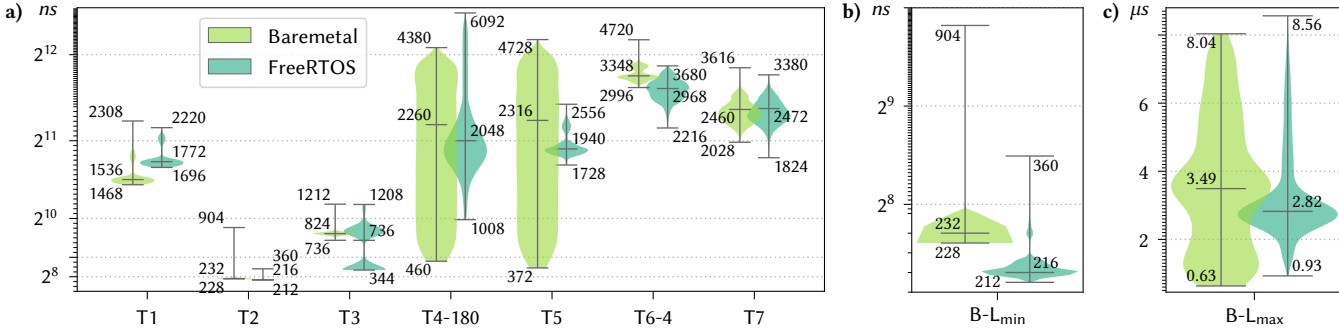

**Figure 4: Latency measured with T1–T7 (a) and B-$L_{min}$ and B-$L_{max}$ (b-c). Figure a) uses a symlog scale with a linear threshold of 2 496 ns, Fig. b) uses a symlog scale with a linear threshold of 240 ns, and Fig. c) uses a linear scale.**

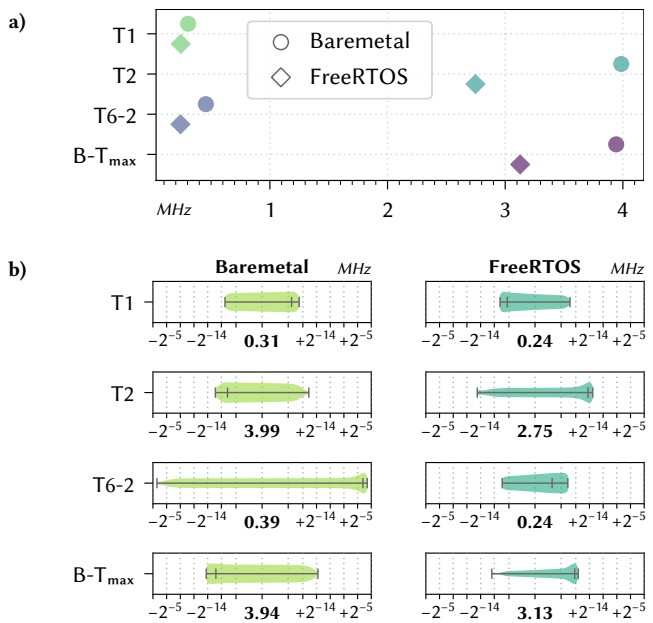

**Figure 5: Throughput measured with T1, T2, T6, and B-$T_{max}$. Figure a) compares the median of all measurements on a linear scale and Fig. b) illustrates the measured throughput ranges on a symlog scale with a linear threshold of 1 Hz, normalized to a 500 kHz range around the highlighted median.**

space constraints we thus only show the results for T6-4, with 4 enabled cores, in case of the latency considerations and T6-2 in case of the throughput measurements.

*T7: Random memory accesses.* As pointed out earlier, the shared memory and interconnecting buses of multi-core processors represent a major source of unforeseen delays. Accordingly, T7 is designed to delay memory accesses by overloading the interconnecting bus and memory interface. For this purpose all 4 cores are running random, concurrent memory accesses in form of constants that are written to random locations in a 96 MB large array. In parallel core 0 executes the standard latency test. Throughput evaluations are not considered with this test-case, as it targets to delay the interrupt handling process.

## 4.2 Proposed Benchmarks

Analyzing the measured interrupt performances under the different test-cases, shown in Figs. 4 and 5, we conclude that first of all different setups and software stacks indeed considerably influence the interrupt handling performance. All three targeted components, provoke a considerable effect on the interrupt latency and throughput. Particularly noticeable are the differences between the test-cases with enabled (T2, T3) and disabled caches (T1, T4–T7), for both the observed latency and throughput, as well as the effects of stressing the GIC on the measured latency (T4–T6).

Of special interest is that the FreeRTOS based stack achieved a smaller minimum latency and a narrower variation range of the latency and throughput, compared to the bare-metal stack. Examples

are for instance T2 and T6-4 for latency measurements, and T6-2 for throughput measurements. After measuring and reviewing the tests for each critical test-case multiple times without finding any anomalies, we assume that some low-level hardware effects, for instance in the pipeline or shared interconnects, might cause the observed behavior. Further insight into the situation could be gained by *(i)* implementing a fully optimized, assembly-only bare-metal stack, or *(ii)* analyzing the actual hardware effects with a cycle-accurate simulation in ARM's Cycle Model Studio [7]. However, both approaches are out of the scope of this paper.

T2 produces by far the shortest interrupt latency of 232 ns on average with only a few outliers. Hence, we propose to utilize T2 as benchmark for the minimal achievable latency (B-$L_{min}$).

To obtain a suitable benchmark for the maximal latency, we analyzed all combination out of the test-cases T4-36, T4-144, T6-3, and T7. Except for the combination out of T6 and T7, all tested combinations showed a similar performance with only slight differences. An exception to that forms the interrupt latency performance of the combination out of T4-144 and T6 on FreeRTOS, which is considerably more confined than all other observed ranges. The highest latency is achieved with a combination out of T4-36 and T6, however, the combination of T4-36, T6, and T7 is close. Accordingly, we propose to use the combination out of T4-36 and T6 to benchmark the achievable maximal interrupt latency (B-$L_{max}$).

For the maximal throughput benchmark (B-$T_{max}$) we evaluated all four variants of the T6 test-case with enabled caches (T2). Interestingly, the enabled caches seem to mitigate the effect of more enabled cores, as all combinations showed a similar throughput. However, the combination out of T6-2 and T2 still performed best. Even though the maximal achieved throughput of the combined test-cases lags a little behind that of T2 alone in case of the bare-metal software stack, it outperforms T2 by far in case of the FreeRTOS based stack. Hence, we propose the combination out of T6-2 and T2 to benchmark the maximal throughput of a system.

## 5 RELATED WORK

In principle, there exist two patented interrupt latency measurement approaches that are used in literature. First, measurements based on an external measurement device, such as an oscilloscope [16]. And second, measurements based on storing timestamps when an interrupt is asserted and when the interrupt handling routine is completed [17], like we do with our measurements.

Liu et al. [18] measured the interrupt latency of five Linux variations on an Intel PXA270 processor, which features an ARM instruction set. They used a counter based measurement method and focused on the effect of different computational loads. Since their stimulation is limited to a single periodic interrupt, we argue that their approach is not able to stress the interrupt distribution process and that they rather analyzed the responsiveness of the scheduler to aperiodic events than the deviation of the interrupt latency.

The wide majority of studies, however, focused on interrupt performance measurements with external measurement devices [19, 22, 26], or combined it with the timestamp approach [24]. Macauley [19] compared different 80×86 processors with each other and NXP Semiconductors [22] determined an exact latency for their i.MX RT1050 processor. All other aforementioned studies focused

on comparing different software stacks with respect to various computational loads. None of the mentioned studies analyzed the throughput, or stressed the interrupt distribution process.

Aichouch et al. [2] claim to have measured the event latency of LITMUSˆRT *vs.* a KVM/Qemu virtualized environment on an Intel based computer. However, it stays unclear how they performed the measurements and where they got the timing information from.

Previous studies of the achievable interrupt throughput focused on the analysis of the achievable network packet transmission/reception or storage input/output operations per second when considering different interrupt coalescing and balancing strategies [1, 8, 23], but do not analyze the interrupt throughput in isolation with respect to different software stacks.

## 6 CONCLUSION AND OUTLOOK

We presented a flexible evaluation method based on the ARM Core-Sight technology [4], which enables the assessment of various software stacks on top of commodity ARMv8-A platforms with respect to their interrupt handling performance. Utilizing the evaluation method, we crafted seven specifically tailored test-cases that were shown to stress the ARM interrupt handling process. Out of these test-cases we deduced three benchmark functions, tailored to provoke the minimal (B-$L_{min}$) and maximal (B-$L_{max}$) interrupt latency, and the maximal throughput (B-$T_{max}$), of a given software stack. We validated the test-cases and benchmark functions by comparing two software stacks (a simple bare-metal and FreeRTOS based environment) and measuring them on top of a Xilinx ZCU102 [30].

Our measurements showed that different software stacks do have a considerable impact on the interrupt handling performance of a hardware platform. Hence, we hope to draw some attention on the importance of a good software design for CPS, with respect to interrupt processing and the need of a more profound analysis on how interrupt handling processes can be made more predictable with respect to the achievable latency and throughput.

## ACKNOWLEDGMENTS

The presented results build on top of the Bachelor's thesis by Wild [29] and were partially funded by the German Federal Ministry of Economics and Technology (BMWi) under grant n°01MD16002C and the European Union (EU) under RIA grant n°825050.

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
