# OpenReview forum: "Quantifying the Latency and Possible Throughput of External Interrupts on Cyber-Physical Systems"
_sigmobile.org/MobiCom/2020/Workshop/CPS-IoTBench — CPS-IoTBench 2020 Conditional_

### Official Review · AnonReviewer4 · 2020-06-28
**really hasn't been done before? also, bad anonymization**

**Rating:** 5
**Confidence:** 3

**Review:**

Positives:
- well written
- in principle, relevant topic
- nicely designed set of benchmarks

Negatives:
- I have a hard time believing that this has never been done before
- results are discussed in very hastily, despite the leftover space on 6th page
- motivation is generality, outcome works only for ARM
- poor anonymization, prevents assessing overlap with previous work

The effort in designing benchmarks for interrupts on embedded systems
is commendable and relevant. Nevertheless, I have a hard time
believing that this has never been done before, although I admit this
is not really my area.

Further, in the intro the authors state that "Previous approaches in
this area require a very detailed knowledge of the utilized hardware
and are therefore not easy to apply" but then end up with a solution
for which (end of sec.3) "The disadvantage of the presented
measurement approach, however, is that it is only applicable for ARM
based platforms." Not sure of the advancement/contribution w.r.t. the
state of the art.


The benchmarks are meaningful, but the discussion of the results is
quite hasty, and mostly relegated in 4.2, whose title should be
renamed more appropriately. I found the results also somewhat
surprising, in particular in Fig.4 it looks like latency is better
with FreeRTOS than bare metal; I would have expected the opposite, due
to the lower overhead.

Finally, the authors implemented anonymization in a way that prevents
the reviewer from assessing overlapping with previous work. Indeed,
instead of citing their work in third person and providing the full
citation to reviewers, at the end of sec.3 the authors mention
"interrupt latency measured in a previous study for the same hardware
platform []". This is unacceptable, as it prevents reviewers from checking the overlap of such work with the submitted one.

---

### Official Review · AnonReviewer3 · 2020-06-28
**Simple idea, unclear takeaways**

**Rating:** 5
**Confidence:** 2

**Review:**

Strengths:
- Simple and intuitive idea for precise measurement of latency and throughput of interrupts
- Most parts are well-written
- Benchmarks seem useful

Weaknesses:
- Related work is quite limited. Has there not been any research on this topic? I could find at least a couple of patents that were measuring latency and throughput without an external system.
- Results are summarized in 1/3 of a page without clear explanation. Benchmarks derived are specific to one processor but how to derive these values across a variety of ARM processors?
- Contradicting claim on contribution: the first sentence of paragraph 3 of Introduction claims to present a flexible interrupt performance measurement for a wide variety of platforms, however, the last paragraph of Section 3.3 rightly says that this approach is limited to ARM based processors.

Details:
- Interesting article based on the intuitive idea of using an external system to measure the latency and throughput of interrupts in ARM based processors. However, there is not much related work that is presented in the paper. Not being from this exact field of research, I see that there are a few patents already on this topic without external measurement device. How would this work compare against measurements without an external system?
- What is the industry standard approach adopted in current CPS to benchmark?
- Results section needs more explanation than provided there. In Fig. 4(a), why is there more deviation in baremetal as compared to FreeRTOS (T1, T2, T5)? This is counter-intuitive.
- The same goes with 4(b) and (c), wherein FreeRTOS gives a much lower benchmark value.
- Is there a more generalized approach to retrieve the benchmarks rather than manual selection?
- Would the results be applicable to ARM GIC architectures v3 and v4, which are the latest ones?

---

### Official Review · AnonReviewer2 · 2020-06-29
**Meaningful direction, narrow appeal**

**Rating:** 6
**Confidence:** 2

**Review:**

Benchmarking interrupt latencies and throughput makes a lot of sense especially when one has to select the best stack/OS/software architecture for designing a real-time system.  The paper, thus, rightly aims to propose a battery of benchmarks for this purpose, although without citing references that sailed in the same boat for similar systems. The proposed procedure exploits built-in platform-specific hardware and software support. Can the proposed benchmarks be extended to other stacks and hardware?


 While discussing results, a further explanation is needed why FreeRToS sometimes outperforms bare-metal stack. FreeRTOS is a popular OS. But there is no adequate information about the "Baremetal". It is not clear whether the higher interrupt latency of Baremetal is due to its unoptimized implementation.


Pro:
+ Meaningful direction on benchmarking interrupt latency performance of stacks that are candidates for designing real-time systems.


Cons:
- Lack of related works focusing on interrupt latency measurement. The earlier work of the author is not referenced.
- Narrow appeal: The evaluation setup and procedure seem very specific to the debug and trace support (CoreSight) and the processor type under test. How this can generalize to other setups is not explained.

---

### Official Review · AnonReviewer1 · 2020-06-29
**Interesting evaluation missing a little bit of detail**

**Rating:** 7
**Confidence:** 4

**Review:**

This paper investigates the performance of interrupt handling on an ARM v8 platform. The paper outlines 7 interrupt-focused test cases, and then benchmarks the Xilinx ZCU102 using both bare-metal firmware and FreeRTOS.

The argument in this paper is interesting, and having a systematic way to evaluate interrupt handling performance on different systems would provide a useful comparison basis when selecting a hardware platform and OS. Having a set of standard benchmarks is really useful for others who want to evaluate the interrupt performance of their systems.

My main concern is about how specific the analysis is to the particular Xilinx chip used in the paper. While the chip has some interesting interrupt handling hardware, could the same test suite and analysis be done on any or most ARM v8 chips? How common is the Coresight tool? When reading the introduction, I immediately thought of the Raspberry PI platform as it is ubiquitous and often a starting point for building an new CPS. Showing even a limited set of the tests on a separate platform would help show the generality of this approach and the test cases.

The paper should discuss test case 5 in more depth. According to Figure 4, bare metal showed a huge range of times, both better and worse than FreeRTOS in this test. I would not expect bare metal to have such a large spread, or to have a worse average and worst case time than running an OS. What accounts for this?

Many of the tests disable caching. Is this representative of a real-world test? Why would users in practice disable caching? I would expect caching to be on by default.

The precision of the Coresight tool is cited as 4 ns, which seems incredibly precise, and largely unnecessary, given that physical systems rarely operate on a timescale anywhere close to that. Providing some context for what measurement precision is needed to get actionable results would be helpful, and would provide guidance to future researchers who want to pursue a similar kind of testing.

---

### Decision · Program_Chairs · 2020-07-07

Conditional Accept